# Cognitive Bias Affects Perception and Decision-Making in Simulated Facial Recognition Searches [note 1]

**DOI:** 10.3390/bs15081094

**Published:** 2025-08-12

**Authors:** Cecelia K. Stewart, Jeff Kukucka

**Affiliations:** Department of Psychology, Towson University, Towson, MD 21252, USA; cstewar1@students.towson.edu

**Keywords:** contextual bias, automation bias, face matching, pattern comparison, forensic science, wrongful convictions, linear sequential unmasking

## Abstract

Cognitive bias can prompt inconsistency and error in visual comparisons of forensic patterns. We tested whether bias can likewise impede attempts to identify unknown criminal perpetrators via facial recognition technology (FRT). Participants (*N* = 149) completed two simulated FRT tasks. In each, they compared a probe image of a perpetrator’s face against three candidate faces that FRT allegedly identified as possible matches. To test for contextual and automation biases, each candidate was randomly paired with either extraneous biographical information or a biometric confidence score, respectively. As predicted, participants rated whichever candidate’s face was paired with guilt-suggestive information or a high confidence score as looking most like the perpetrator’s face, even though those details were assigned at random. Furthermore, candidates randomly paired with guilt-suggestive information were most often misidentified as the perpetrator. These findings indicate a clear need for procedural safeguards against cognitive bias when using FRT in criminal investigations.

## 1. Introduction

Criminal investigations often rely on forensic pattern comparisons, wherein trained examiners visually compare two items (e.g., fingerprints, bullets)—typically one of unknown origin (e.g., a bullet found at a crime scene) and one of known origin (e.g., a bullet from a suspect’s gun)—and render a judgment as to whether they share a common source (i.e., “match”). While people tend to believe that these judgments are highly scientific and practically infallible ([8]; [23]), false or misleading forensic evidence has been implicated in over 1000 wrongful criminal convictions ([35]), leading scholars to investigate why those errors occurred and how they can be prevented.

There is now ample research evidence that forensic pattern comparisons are susceptible to *cognitive bias*—i.e., the natural tendency for a person’s beliefs, expectations, motives, and/or situational context to influence their perception and decision-making ([26]; [46]). Simply put, examiners with different mindsets or procedures may reach different judgments of the very same evidence, which increases the risk of error insofar as one of those differing judgments must be incorrect. Below, we first explain two common sources of cognitive bias in forensic pattern comparisons. Then, we discuss the growing use of facial recognition technology in criminal investigations, and we argue that it may be likewise vulnerable to cognitive bias. Finally, we describe an experiment that tested that possibility.

### 1.1. Contextual Bias

Ideally, forensic decisions should reflect a professional’s unique expertise as applied to the information in question. However, it is now well understood that cognitive biases can affect a wide range of forensic decisions, including not only visual comparisons of forensic patterns but also forensic psychiatric evaluations ([4]; [36]) and autopsy outcomes ([13], [15]), among others.

In the forensic sciences, for example, examiners often receive additional contextual information—such as details about a suspect’s prior legal involvement ([18])—that should have no bearing on their judgment. *Contextual bias* occurs when that extraneous information nonetheless inappropriately affects an examiner’s judgment.

In an early demonstration, [10] ([10]) found that fingerprint examiners changed 17% of their own prior judgments of the same prints after being led to believe that the suspect had either confessed or provided a verified alibi, information that implies that the two prints should or should not “match”, respectively. In another study, DNA analysts formed different opinions of the same DNA mixture if they knew that one of the suspects had accepted a plea bargain ([11]). Similar contextual bias effects have been replicated across myriad forensic disciplines (e.g., toxicology, anthropology, bloodstain pattern analysis, digital forensics; see [26]): in each case, examiners’ knowledge of irrelevant contextual information distorted their judgments of the items in question. In response, the [34] ([34]) has urged examiners to “draw conclusions solely from the physical evidence… and not from any other evidence in the case.”

Importantly, contextual bias is especially prone to affect judgments that are difficult or ambiguous. For instance, studies have found that extraneous case information had a stronger biasing effect on examiners’ judgments of “difficult” rather than “not difficult” fingerprints ([10]), distorted or incomplete rather than pristine bitemarks ([39]), and inconclusive rather than conclusive polygraph charts ([16]).

### 1.2. Automation Bias

In some forensic disciplines, examiners have access to technologies that can examine an unknown item, search it against a large database of known items, and produce a list of potential “matches”. However, the final decision is still made by a human examiner. *Automation bias* occurs when the examiner is overly reliant on metrics generated by the technology such that the technology usurps rather than supplements their judgment ([40]).

For instance, fingerprint examiners can enter an unknown print into the Automated Fingerprint Identification System (AFIS), which then searches a database of known prints and returns a rank-ordered list of known prints that AFIS’s algorithm judged as highly similar to the unknown print. The examiner then decides which, if any, of the search results “matches” the unknown print, which is a very difficult task insofar as the search results often include “close non-matches” (i.e., prints that are highly similar to the unknown print, but actually belong to a different person), which pose a greater risk of false identification ([24]).

To test whether the AFIS’s rank ordering invites automation bias, [14] ([14]) performed AFIS searches of unknown fingerprints, but they randomized the order of the search results before giving them to examiners to see if they would be biased toward whichever print the algorithm ostensibly judged as most similar. Sure enough, examiners spent more time analyzing whichever print just so happened to appear at the top of the list, and they more often identified that print as a “match” to the unknown print, regardless of whether it actually was. In response, some examiners have advocated “to remove the score and shuffle the candidate list for comparison” so that those metrics cannot influence examiners’ judgments ([19]).

### 1.3. Facial Recognition Technology

When an unknown person commits a crime and is captured on camera, it is increasingly common for investigators to use facial recognition technology (FRT) to attempt to identify the unknown perpetrator ([1]). Similarly to AFIS, FRT takes an image of an unknown person (i.e., a *probe* image; e.g., from a surveillance camera), compares it against a large database of known faces (e.g., mugshots, driver’s license and passport photos, social media photos), and returns a list of potential “matches” to the unknown face (i.e., *candidate* images). An examiner then scrutinizes the list of candidates and decides which, if any, depicts the same person as the probe image and therefore presumably committed the crime in question.

These judgments are very difficult: On simulated FRT tasks, even professional facial examiners have shown mean error rates around 30%, with individual examiners varying widely in accuracy ([51]; [58]). Moreover, the images used in those studies were of higher quality than the probe images typically used in criminal investigations, which tend to be blurry, poorly lit, and/or only show part of the person’s face, which can diminish accuracy even further ([2]; [33]). Finally, it is unclear whether training can meaningfully improve face matching ability (e.g., [52], [53]), but in any case, many agencies do not require that FRT users be trained in facial recognition, and consequently few are ([54]).

Compounding these issues, many FRT systems expose users to known risk factors for cognitive bias by providing for each candidate image (a) any known biographical information about that candidate (e.g., prior legal involvement), which may prompt contextual bias, and (b) a score that quantifies the system’s confidence that the candidate and probe images depict the same person, which may prompt automation bias ([1]). As noted above, these factors have been shown to bias expert judgments in other forensic domains, and it thus stands to reason that they could likewise bias judgments of FRT search results ([29]), especially given the aforementioned inherent difficulty of such judgments.

Of note, at least two prior studies have shown that extraneous contextual information can distort judgments of facial similarity. First, [3] ([3]) found that people who viewed photos of adult–child pairs judged the adult and child as more facially similar if led to believe that the two were genetically related (e.g., mother and son), regardless of their actual relatedness. In fact, beliefs about relatedness had a stronger effect on judgments than did actual relatedness. Second, in a forensic context, [7] ([7]) found that mock investigators who saw photos of four criminal suspects judged whichever suspect had allegedly been implicated by two eyewitnesses as looking most similar to a computerized facial composite of the perpetrator, even though that suspect was actually chosen at random.

### 1.4. The Current Study

The current study was designed to test whether contextual bias and/or automation bias can distort judgments of FRT search results and thus impede attempts to identify unknown criminal perpetrators. Participants acting as mock forensic facial examiners completed two simulated FRT tasks, each of which included a probe image of a criminal perpetrator and three candidate images that FRT allegedly identified as potential “matches”. To test for automation bias, one FRT task randomly assigned a high, medium, or low numerical confidence score to each candidate. To test for contextual bias, the other FRT task randomly assigned extraneous biographical information to each candidate—namely that they had committed similar crimes in the past, were already incarcerated when this crime occurred, or had served in the military (control). Participants then separately rated each candidate’s similarity to the probe and indicated which, if any, of the three candidates they believed was the same person depicted in the probe image (and therefore committed the crime).

**H1.** 
*Automation bias will affect participants’ FRT judgments such that they will rate whichever candidate is randomly assigned a high confidence score as looking most similar to the probe, and they will most often misjudge that candidate as the perpetrator. Conversely, they will rate whichever candidate is randomly assigned a low confidence score as looking least similar to the probe, and they will least often misjudge that candidate as the perpetrator.*


**H2.** 
*Contextual bias will affect participants’ FRT judgments such that they will rate whichever candidate allegedly committed similar crimes in the past as looking most similar to the probe, and they will most often misjudge that candidate as the perpetrator. Conversely, they will rate whichever candidate was allegedly incarcerated when the crime occurred as looking least similar to the probe, and they will least often misjudge that candidate as the perpetrator.*


## 2. Method

### 2.1. Participants

Participants were recruited online via Prolific and each paid USD 2.65. Recruitment was limited to adults (age 18+) residing in the United States, and participants were required to complete the study on a desktop or laptop computer (i.e., not a cell phone or tablet) to ensure an adequate view of the study materials. There were no other eligibility criteria.

An a priori power analysis using G*Power 3.1.9.7 ([17]) indicated a required sample size of *N* = 116 for 95% power to detect medium effects (f = 0.15) in a one-way repeated-measures design with three levels. Because we anticipated needing to later exclude some low-quality responses ([9]), we oversampled and recruited *N* = 156 US-based Prolific users. We later excluded data from seven participants (4.5%) who correctly guessed the purpose of the study, leaving a final sample of *N* = 149. However, one participant completed only one of the two simulated FRT tasks (i.e., the context task), so analyses for that task are based on *N* = 148.

Our final sample had a mean age of 36.41 years (*SD* = 11.31; range = 19–76) and similar numbers of men (47.7%) and women (51.0%), plus one non-binary person and one person who declined to report their gender. Most participants self-identified as White (69.1%), with fewer self-identifying as Black (19.5%), Asian (8.1%), or another race (3.4%).

### 2.2. Procedure

Participants volunteered for the study via Prolific, where it was advertised as a study to compare the efficacy of two different facial recognition programs. They then completed the study via Qualtrics. After giving consent, they read instructions that explained how facial recognition technology is used in criminal investigations and described the tasks that they would be asked to complete. The full text of these instructions is available on OSF (https://osf.io/fz9qn/, accessed on 25 July 2025).

Each participant then completed two facial recognition tasks in a counterbalanced order. For each task, they saw a probe image of a criminal perpetrator along with a short description of the crime and three candidate images that the facial recognition program had allegedly identified as possible “matches” to the unknown perpetrator. All three candidate images were shown on the same webpage, and each was shown alongside the probe image, such that participants separately compared each candidate against the probe for as much time as they desired and separately rated each candidate’s similarity to the probe. Lastly, participants indicated which, if any, of the three candidates they believed “matched” the probe image and therefore committed the crime.

For one of these tasks (i.e., the *automation* task), the probe and candidates were all White men, the crime was armed robbery, and each candidate was accompanied by a “biometric confidence score” that the facial recognition program had allegedly calculated as a measure of its similarity to the probe (i.e., 0–100, with higher scores indicating greater perceived similarity). The three candidates were always given biometric scores of 98, 90, and 84, respectively, but the three candidates’ faces were shown in a random order for each participant.

For the other task (i.e., the *context* task), the probe and candidates were all Black men, the crime was robbery, and each candidate was accompanied by extraneous biographical information about that person, which suggested either guilt (i.e., previously convicted of similar crimes), innocence (i.e., was already in prison when this crime occurred), or neither (i.e., discharged from the military). The three candidates’ faces were always shown in the same order, but the three types of contextual information were shown in a random order for each participant.

After completing both tasks, participants self-reported their age, gender, and race, completed suspicion and attention checks, and were debriefed on the purpose of the study.

### 2.3. Materials

#### 2.3.1. Probe and Candidate Images

First, we located three probe images from actual criminal investigations, each of which was publicly available online and depicted a criminal suspect’s face as captured by a surveillance camera. Next, we performed reverse image searches (via Google) for each of those three probes in order to identify publicly available images (e.g., from social media) of similar-looking (but presumably innocent) individuals who could potentially serve as candidates. From this, we were able to identify between five and nine potential candidates for each probe.

Then, we pilot-tested these images so that we could select eight of them (i.e., two probes each with three candidates) for use in the main study. Undergraduate students (*N* = 91) viewed and rated 21 pairs of images in a randomized order, with each pair including one of the three probes and one potential candidate for that probe. For each pair, participants reported (a) how similar the two faces appeared, on a scale from 1 (*not at all similar*) to 7 (*extremely similar*) and (b) whether they believed that the two images could be photos of the same person taken at different times (yes/no). Based on the pilot data, we eliminated one probe for which only two of the nine potential candidates had a mean rating above 2.54. For the remaining two probes, we selected the three potential candidates with the highest mean similarity ratings, which ranged from 2.95 to 3.73 for one probe (a White man) and from 2.69 to 3.07 for the other probe (a Black man). Our pilot data and all eight images used in the study are available on OSF (https://osf.io/fz9qn/).

#### 2.3.2. Simulated User Interfaces

To further our cover story that the study was comparing two different facial recognition programs, we created two simulated user interfaces in which to display the probe and candidate images, and we led participants to believe that they were viewing screenshots from those programs. The user interface for the *automation* task was a black background with the probe and candidate images on the right and a gray text field on the left, which displayed the candidate’s name and social security number (both redacted), their randomly assigned biometric score (explained below), and a list of specific facial features that were compared. The user interface for the *context* task was a gray background with the probe and candidate images on the right and a gray text field on the left, which displayed the candidate’s name and social security number (both redacted), their birth date and height, and their randomly assigned contextual information (explained below). The actual images shown to participants are available on OSF (https://osf.io/fz9qn/).

#### 2.3.3. Automation Task

For the *automation* task, each of the three candidate images was presented along with a “biometric confidence score”, and participants were told that “higher scores mean that the algorithm rated it as more similar to the probe”, but in actuality, the scores were assigned at random. To be exact, all participants saw the same probe image (a White man) and the same three candidate images in a randomized order, with the first, second, and third candidates always having biometric confidence scores of 98, 90, and 84, respectively.

#### 2.3.4. Context Task

For the *context* task, each of the three candidate images was presented along with “any known information about the person”. All participants saw the same probe image (a Black man) and the same three candidate images in the same order, but each candidate was randomly assigned one of three forms of extraneous biographical information: one was said to have been convicted of similar crimes in the past and recently paroled (*guilty* expectation), one was said to have already been incarcerated during the crime in question (*innocent* expectation), and the third was said to have served in the U.S. Coast Guard (*neutral* expectation).

### 2.4. Measures

#### 2.4.1. Dependent Measures

For each task, participants were able to compare the probe and candidate images for as much time as they desired, and they separately rated the degree of similarity between the probe image and each of the three candidate images on a scale from 1 (*not at all similar*) to 7 (*very similar*). Then, participants indicated which of the three candidates they believed had committed the crime by either selecting one of the candidate’s photos or selecting “none of them”.

#### 2.4.2. Suspicion and Attention Checks

After completing both tasks, participants were asked to briefly explain the purpose of the study in an open-ended fashion. Seven participants (4.5%) were later judged to have correctly inferred the true purpose of the study (e.g., “if the information provided on each person would influence the decision”) and so their data were excluded from all analyses.

Participants also completed an attention check, for which we gave them six statements about the candidates they saw earlier and asked them to select only the statements that were true. In actuality, four of the statements were true (e.g., “One person served in the US Coast Guard.”) and two were false (e.g., “One person had a biometric score of 66.”). We intended to exclude participants who misjudged more than one of these statements as true or false, but participants unexpectedly performed very poorly on this attention check: only 21 participants (14.1%) made fewer than two errors. Rather than changing our exclusion criterion after the fact, we opted to retain all 149 participants for analysis regardless of their attention check score.

## 3. Results

Raw data are available on OSF (https://osf.io/fz9qn/). For the automation and context tasks, respectively, Figure 1 and Figure 2 show participants’ mean similarity ratings for each candidate, as well as the percentage of participants who misidentified each candidate as the perpetrator.

### 3.1. Automation Task

Supporting H1, a one-way repeated-measure ANOVA revealed that the randomly assigned biometric confidence scores affected participants’ similarity ratings of the candidate images, *F*(2,296) = 15.84, *p* < 0.001, η^2^_p_ = 0.10. Simple planned contrasts (with the middle confidence score of 90 as the reference category) indicated that participants rated whichever candidate was assigned a score of 98 (*M* = 4.21, *SD* = 1.60) as looking more similar to the probe than whichever candidate was assigned a score of 90 (*M* = 3.58, *SD* = 1.58), *p* < 0.001, and they rated whichever candidate was assigned a score of 84 (*M* = 3.20, *SD* = 1.72) as looking less similar to the probe than whichever candidate was assigned a score of 90, *p* = 0.029.

For this task, 37 participants (24.8%) believed that none of the three candidates was the same person depicted in the probe, and two participants (1.3%) did not answer this question. For the 110 participants who identified one of the candidates as the perpetrator, we performed a one-way chi-squared test to see if their decisions were evenly distributed across the three candidates, which they were: χ^2^(2) = 0.13, *p* = 0.938, *V* = 0.03. In other words, contrary to H1, confidence scores did not affect identification decisions: candidates with scores of 98 (34.5%), 90 (31.8%), and 84 (33.6%) were equally likely to be misidentified as the perpetrator.

### 3.2. Context Task

To evaluate H2, a one-way repeated-measure ANOVA revealed that extraneous contextual information had a non-significant, but marginal effect on participants’ similarity ratings of the candidate images, *F*(2,294) = 2.88, *p* = 0.058, η^2^_p_ = 0.02. To test the specific predictions within H2, we also performed simple planned contrasts (with military service as the reference category), which indicated that participants rated whichever candidate allegedly committed similar crimes in the past (*M* = 3.53, *SD* = 1.77) as looking more similar to the probe than whichever candidate allegedly served in the military (*M* = 3.14, *SD* = 1.71), *p* = 0.028, thus providing partial support for H2. However, contrary to H2, they rated the candidate who was already incarcerated when the crime occurred (*M* = 3.22, *SD* = 1.49) no differently from the candidate who served in the military, *p* = 0.644.

For this task, 65 participants (43.6%) believed that none of the three candidates was the same person depicted in the probe. For the 84 participants who identified one of the candidates as the perpetrator, we again used a one-way chi-squared test to see if their decisions were evenly distributed across the three candidates, which they were not: χ^2^(2) = 12.21, *p* = 0.002, *V* = 0.38. Supporting H2, whichever candidate allegedly committed similar crimes in the past was more often misidentified as the perpetrator (51.2%) than the candidate who was incarcerated during the crime (26.2%) or the candidate who served in the military (22.6%).

## 4. Discussion

As predicted, both contextual and automation biases impeded simulated attempts to use facial recognition technology (FRT) to identify unknown criminal perpetrators. As for contextual bias, participants who saw three possible “matches” to an unknown perpetrator’s face judged whichever person had allegedly committed similar crimes in the past as looking more similar to the perpetrator and more often misidentified that innocent person as the perpetrator, even though that information was actually random. As for automation bias, participants judged possible “matches” as looking more similar to the perpetrator if they were told that the technology had detected greater similarity between them (and vice versa), even though that information was again random. Our results thus corroborate findings from other forensic disciplines (e.g., [14]; [26]) by providing clear evidence that extraneous information—whether from external sources or from the technology itself—can inappropriately influence judgments of FRT search results in ways that increase the risk of costly errors. Indeed, because all candidate images in the current study depicted presumably innocent people, anyone who identified *any* of those candidates as the perpetrator (74% for the automation task and 57% for the context task) made an error that wrongly incriminated an innocent person.

Moreover, there are at least three reasons to believe that our data underestimate the magnitude of these biases. First, our candidates were not terribly similar to our probes: across the pilot and main studies, only one candidate image produced a mean similarity rating above the scale midpoint. In contrast, FRT systems search a much larger database and thus present more “close non-matches” (i.e., extremely similar, but non-matching faces) that are more conducive to bias ([28]) and to wrongly implicating an innocent person ([24]). Second, we showed each participant only three candidates, but FRT systems often return dozens or even hundreds of candidates, and longer lists have been shown to reduce identification accuracy ([21]). Third, we did not exclude participants who performed poorly on attention and manipulation checks, which tends to weaken treatment effects ([32]).

These findings beg the question of how FRT users can and should protect themselves against cognitive bias. Broadly, FRT users stand to benefit from an approach called *Linear Sequential Unmasking—Expanded* (LSU-E; [12]), which provides a general framework for exposing and protecting against biased decision-making and has already been applied to a wide range of domains (e.g., forensic science analyses, workplace safety inspections, deepfake image detection; see [27]). Under LSU-E, decision-makers should avoid information that is irrelevant to the decision, whereas information that is relevant to the decision, but also potentially biasing should be minimized, delayed, and documented.

The most cautious approach would be for FRT systems to altogether eschew any potentially biasing contextual information about the individuals depicted in the candidate images (e.g., prior legal system involvement), hide any computer-generated metrics from FRT users, and randomize the rank order of candidate images—as has been proposed for fingerprint identifications using the AFIS ([19]). Such an approach would maximize confidence that FRT users’ judgments were based on the candidates’ faces rather than swayed by extraneous factors. Alternatively, those metrics could remain hidden from FRT users until after they have reached and recorded a decision, at which point that information could be revealed and the user could adjust their prior decision as desired, provided that any such changes were clearly documented (see [42]). Such an approach would not necessarily prevent bias, but it would make its potential influence more transparent—e.g., if a user’s judgment of the same face changes dramatically after being exposed to computer-generated metrics. As in other domains of forensic pattern comparison, all FRT judgments should ideally be verified by a qualified peer who is kept unaware of the original user’s judgment (i.e., blind verification).

Insofar as FRT candidate lists resemble eyewitness lineups, best practices for FRT use can also benefit from the vast literature on best practices for eyewitness identification (see [56]). For example, FRT users should be reminded that the candidate list may not include the perpetrator ([48]) and then view candidates one at a time to discourage them from simply choosing whichever candidate most resembles the probe ([47]). However, a key difference between eyewitness lineups and FRT candidate lists is that the former embed a suspect’s photo among known-innocent “filler” photos that are known to be erroneous if selected, whereas the latter functions as an all-suspect lineup, which has been likened to “a multiple-choice test with no wrong answer” ([59]). This could be addressed by including known-innocent faces in FRT candidate lists (e.g., similar-looking persons who were deceased when the crime occurred), which has also been proposed in other pattern comparison disciplines as a way to mitigate bias, estimate error rates, and expose unreliable techniques or examiners ([43]; [57]).

Two unexpected findings also emerged. First, contextual information that suggested guilt increased both similarity ratings and misidentifications as predicted, but information that implied innocence did not decrease those same metrics. We suspect this is because the latter manipulation was more subtle and therefore may have been lost on inattentive participants. That is to say, some participants may have noticed that the “innocent” candidate was previously incarcerated without noticing that he was incarcerated at a time that made it impossible for him to have committed the crime, which may have instead created an expectation of guilt—the opposite of what was intended.

Second, contextual information affected identification decisions, but confidence scores did not affect identification decisions. However, the race of the probe and candidates was confounded with the nature of the biasing information—i.e., the context task always featured Black faces, and the automation task always featured White faces. Given that our participants were predominantly White (69%), that cross-race identifications are more difficult than same-race identifications ([31]), and that more difficult judgments are more vulnerable to cognitive bias ([28]), it stands to reason that cross-race FRT judgments may be more vulnerable to cognitive bias, which may explain the aforementioned discrepancy. Alternatively, it may be that narrative information has a stronger biasing effect than numerical information or that there is an interactive effect between race and information type. Because the design of the current study did not allow us to differentiate between these possibilities, future research should investigate whether and how race moderates cognitive bias effects on FRT judgments.

It should be reiterated that our participants were not professional facial examiners, but rather community members with presumably no formal training and little experience with facial comparisons, which may raise questions about the extent to which our findings generalize to real-world FRT users. As for training, research suggests that its benefits are small and inconsistent ([52]), and in any event, real-world FRT users rarely have formal training in facial comparison. For instance, a recent report found that only 5% of FBI agents who used FRT had completed formal training, and across seven U.S. government agencies, nearly 60,000 FRT searches had been conducted by untrained users ([54]). As for experience, real-world FRT users surely have more than our participants, who completed only two simulated FRT trials. However, experience can only improve accuracy to the extent that users also receive feedback on the correctness of their decisions, which enables them to refine their approach over time ([44]). On the other hand, there is some evidence that professional facial examiners tend to make more conservative decisions than novices ([41]; [45]), which would grant some protection against erroneous and/or overconfident decisions.

More broadly, it is critical to note that *accuracy* and *bias* are distinct issues, and there is little reason to believe that training and experience beget immunity to bias, given that bias tends to affect decision-making automatically and outside of conscious awareness ([22]; [37]). In the only existing study to directly compare the proneness to bias of forensic experts and novices, [55] ([55]) found that biasing information had an equivalent impact on experienced crime scene investigators’ and undergraduate students’ impressions of and behavior at a mock crime scene. Some have argued that expertise can even increase vulnerability to bias inasmuch as experts make greater use of heuristics and other cognitive shortcuts ([5]). In any case, future research should examine whether training and experience diminish (or exacerbate) vulnerability to cognitive bias among FRT users specifically.

Relatedly, the current study featured only target-absent trials, which precludes a full assessment of how biasing information affects accuracy. Some have noted that reliance on contextual information can enhance decision accuracy (e.g., [6]; [30]), but such an argument is dangerously misguided, because bias can “lead forensic examiners to the right decision, but for the wrong reasons” ([25]). For instance, if an FRT user bases a decision on irrelevant contextual information and that decision happens to be correct, it still carries no value because the decision did not stem from their unique expertise, yet factfinders are likely to assume otherwise, give the examiner’s decision more weight than it merits, and perhaps also inappropriately double-count evidence as a result (see also [50]). With these important caveats, future studies of bias in FRT decisions should nonetheless include both target-absent and target-present candidate lists to provide a fuller picture of how bias may produce both false-positive and false-negative errors.

In sum, despite FRT’s widespread use and known contributions to wrongful arrests—which both harm innocent individuals and enable perpetrators to victimize others (e.g., [20]; [38])—there remains surprisingly little regulation or standardization of how it can and should be deployed in criminal investigations ([1]). Accordingly, the [33] ([33]) recently called for more “research to improve the accuracy [and] further explore the sociotechnical dimensions of current and potential [FRT] uses” (p. 125). To that end, our findings indicate a clear need for procedural safeguards against cognitive bias in FRT judgments as in other forensic pattern comparison disciplines.

## Figures and Tables

**Figure 1 behavsci-15-01094-f001:**
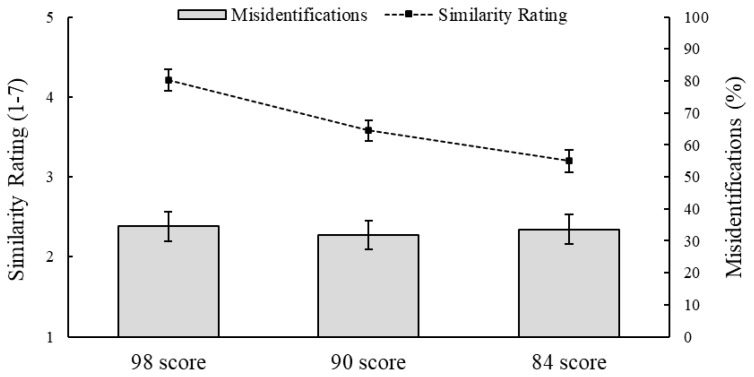
Effects of biometric confidence score on similarity ratings and misidentifications in the automation task. *Note*. Mean similarity ratings are based on *N* = 149. Misidentification percentages are based on *n* = 110 (73.8%) who identified one of the three candidates as the perpetrator.

**Figure 2 behavsci-15-01094-f002:**
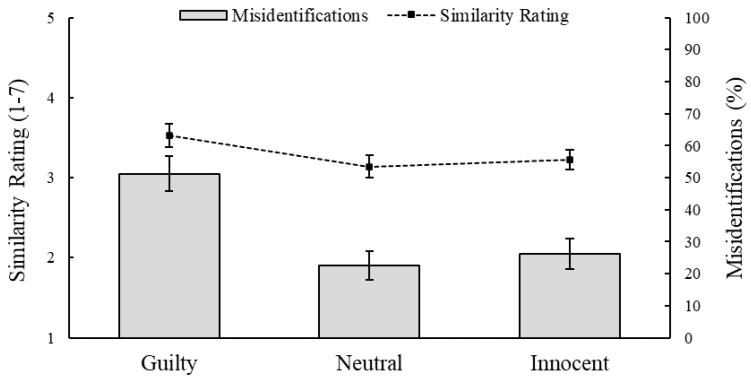
Effects of extraneous biographical information on similarity ratings and misidentifications in the context task. *Note*. Mean similarity ratings are based on *N* = 148. Misidentification percentages are based on *n* = 84 (56.8%) who identified one of the three candidates as the perpetrator.

## Data Availability

All study materials and data are available on the Open Science Framework (https://osf.io/fz9qn/).

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
