# Peer review of "Cognitive Bias Affects Perception and Decision-Making in Simulated Facial Recognition Searchesâ€"

_behavsci, 2025, doi:10.3390/bs15081094_

Round 1
Reviewer 1 Report
Comments and Suggestions for Authors
Summary of the Paper
This paper reports a study examining how information from face recognition technology (FRT) influences human face matching decisions in forensic contexts. In two tasks, participants were presented with a single probe face and three non-matching candidate faces. In one task, candidate faces were accompanied by automated similarity scores from FRT; in the other, by contextual information (e.g. prior criminal records). Participants rated the similarity of each candidate to the probe and judged whether any matched the probe. The authors conclude that both FRT scores and contextual cues influenced participants’ decisions, with higher similarity scores and incriminating context increasing perceived similarity. Contextual information also led to a higher number of misidentifications. They interpret these findings as evidence that both forms of information can bias face-matching decisions in forensic contexts.
The paper is clearly written, and the topic is timely and relevant. However, several aspects of the study design—such as the use of untrained participants, the single-trial format, and the inclusion of participants who failed attention checks—limit the extent to which the findings can be generalised to applied forensic settings. One of the key findings reported (the influence of contextual information on similarity ratings) was also not statistically significant, despite being interpreted as if it were. While the study raises relevant questions and has the potential to contribute to ongoing discussions about FRT in forensic contexts, While the study raises relevant questions and has the potential to contribute to ongoing discussions about FRT in forensic contexts, I feel that substantial revisions and a more cautious interpretation of the findings would strengthen the paper and help it better support its intended contribution.
Main Comments
Relevance and Scope
The topic of this study is timely and relevant, particularly given the growing use of face recognition technology (FRT) in security and forensic contexts. The authors aim to examine how FRT outputs and contextual information influence human face matching decisions—a worthwhile and important question. The paper is clearly written and generally well-structured, which aids comprehension of the design and results. However, several aspects of the study design limit the extent to which the findings can be generalised to real-world forensic practice.
Applicability to Forensic Contexts
A key concern is the mismatch between the participant sample and the professional roles the study appears to reference. The sample consisted entirely of untrained participants, whereas applied face matching is typically carried out by professionals with extensive specialist training (e.g., forensic examiners or reviewers) or individuals with enhanced natural abilities (e.g., super recognisers). This gap significantly limits the practical implications of the findings and should be more explicitly acknowledged.
Moreover, the manuscript frequently refers to “facial examiners,” but forensic facial examiners typically work methodically, making meticulous feature-by-feature comparisons, often over extended periods and sometimes collaboratively. The tasks used in this study, which involve rapid, intuitive decisions, therefore do not reflect the comparisons conducted by forensic examiners. They are more akin to the types of judgements that passport officers are required to make. Clarifying the target population for the research would help situate the study more appropriately within the existing literature.
Additionally, there is evidence that forensic examiners make more cautious decisions than untrained controls and are less likely to make high-confidence errors (Phillips et al., 2018; Sexton et al., 2024). This raises further concerns about generalisability, as the influence of FRT scores and contextual cues on untrained participants may not reflect how trained professionals would behave.
Study Design and Methodological Constraints
The decision to include only target-absent trials is a major limitation. While this design avoids issues of bias due to known matches, it also excludes scenarios where target-present lineups are common—an important part of real forensic practice. The omission of target-present trials should be discussed more clearly, particularly as the role of contextual or automated information might differ under those conditions.
The single-trial design also limits the conclusions that can be drawn. Participants had no opportunity to adjust strategies, calibrate to the task, or develop a sense of the reliability of the FRT output. In applied settings, professionals work with repeated exposure and are likely to develop a sense of when and how to rely on technological tools. Notably, the authors do not acknowledge the implications of the single-trial format, which is an oversight.
Participant Screening and Attention
It is unclear whether participants with conditions affecting face perception (e.g., prosopagnosia, autism, or visual impairments) were excluded. It is common in face matching research to screen for such factors, given their potential impact on performance. Clarification on this point would improve transparency.
More concerningly, the authors state that participants who failed attention checks were not excluded. This is problematic, as forensic judgments are typically made in high-stakes, focused conditions. Including inattentive or disengaged participants risks introducing noise, undermining confidence in the results.
Procedural Details
It would be helpful to specify how long the stimuli were presented to participants, and whether the probe and candidate faces were shown simultaneously or sequentially. Including a diagram of the trial structure would clarify the procedure.
Ethnicity and Cross-Race Effects
The use of faces from different ethnic backgrounds in the two tasks raises the possibility of a cross-race effect influencing performance, particularly as the context task featured only Black faces and the sample was predominantly White. The increased number of misidentifications in this task may be attributable at least in part to this factor. The authors mention this briefly in the discussion, but it deserves more thorough consideration, especially given the observed differences in performance across the two tasks.
Statistical Reporting
Some results are described as “marginally significant” with a p-value of 0.058, which does not meet the standard significance threshold. These results should be reported accurately.
Reviewer 2 Report
Comments and Suggestions for Authors
I thank the authors for the opportunity to review this timely and well-conceived manuscript, which explores how contextual and automation biases may influence face-matching decisions in simulated facial recognition tasks.
The study addresses a critical and underexamined intersection between human judgment and algorithmically assisted forensic tools. The design is overall well executed, and the findings are clearly reported. That said, a few points may help strengthen the manuscript. First, while the introduction focuses appropriately on cognitive bias in forensic pattern comparison, it would benefit from a slightly broader conceptual framing. Biases in forensic decision-making are well documented not only in visual comparisons but across various domains, including expert interpretation, witness evaluation, and mental state assessment. A more inclusive overview would help to position the present study as part of a wider concern around cognitive vulnerability in forensic practice. In this regard, the authors may consider citing recent work in forensic psychiatry that outlines common cognitive distortions in evaluations of criminal responsibility and risk. For example, Buongiorno et al. (2025) provide an updated synthesis of bias mechanisms in forensic psychiatric settings, which may serve to underscore the pervasiveness of these processes across distinct but related domains (https://doi.org/10.1016/j.ijlp.2025.102083). Second, the methodological choice to assign White faces to the automation task and Black faces to the contextual task introduces a potential confound linked to the cross-race effect. This issue is briefly acknowledged in the discussion but would merit further consideration or statistical control. Third, the similarity between probe and candidate images appears to have been relatively low based on pilot data. In real-world FRT settings, close non-matches are a major source of error, and the relatively low visual similarity in the present tasks may have reduced the ecological challenge and possibly attenuated the biasing effects. Fourth, although the authors initially planned to exclude participants who failed attention checks, they ultimately retained them without conducting sensitivity analyses. Including a brief robustness check—even in supplementary materials—would improve transparency and reinforce the validity of the findings. Additionally, the use of ANOVA and chi-square tests is adequate, but given the repeated-measures design and categorical outcome variables, a mixed-effects model or logistic regression could provide a more appropriate analytic framework, especially for identification decisions. The finding that confidence scores influenced similarity ratings but not identification outcomes, in contrast to contextual information, raises interesting questions about how different types of data are processed and prioritized by users. Expanding on this asymmetry—perhaps by referencing literature on narrative versus numerical information processing—could enrich the theoretical discussion. Finally, while the recommendations regarding Linear Sequential Unmasking-Expanded (LSU-E) are welcome, they would be more actionable if tied to concrete examples of how such protocols could be applied in practice—for instance, by masking or delaying access to certain information during FRT review. Overall, the manuscript is a valuable contribution, and with modest refinements, it has the potential to meaningfully inform both academic understanding and applied forensic procedures.
Round 2
Reviewer 2 Report
Comments and Suggestions for Authors
Thank you for your thoughtful and detailed revisions. The manuscript has substantially improved, and several additions—such as the robustness checks and the discussion of LSU-E—are commendable.
However, two critical issues remain:
- Conceptual scope: The Introduction still omits any mention of cognitive biases in adjacent forensic domains, particularly forensic psychiatry. While you argue that this falls outside the scope of your study, even a brief reference to well-documented bias mechanisms in psychiatric or psychological forensic evaluations (e.g., Buongiorno et al., 2025) would help frame your work within the broader literature on cognitive vulnerability in forensic decision-making. This would not detract from your specific focus, but rather reinforce its relevance.
- Analytical approach: While I understand your preference for ANOVA and chi-square tests, the repeated-measures structure and categorical outcome variables make mixed-effects models or logistic regression more appropriate. Reader familiarity should not override statistical rigor, especially when more suitable methods are readily available and interpretable.
These two points should be addressed to meet the standards of analytic and conceptual completeness expected for publication.
Round 3
Reviewer 2 Report
Comments and Suggestions for Authors
Dear Authors,
Thank you for incorporating the suggested citation. As for the statistical approach, I note your decision to retain the original analyses; while alternative models might have offered additional nuance, your rationale is understood.
Kind regards